# Assessment of Body Composition and Physical Performance of Young Soccer Players: Differences According to the Competitive Level

**DOI:** 10.3390/biology11060823

**Published:** 2022-05-27

**Authors:** Stefania Toselli, Mario Mauro, Alessia Grigoletto, Stefania Cataldi, Luca Benedetti, Gianni Nanni, Riccardo Di Miceli, Paolo Aiello, Davide Gallamini, Francesco Fischetti, Gianpiero Greco

**Affiliations:** 1Department of Biomedical and Neuromotor Sciences, University of Bologna, 40126 Bologna, Italy; stefania.toselli@unibo.it; 2Department for Life Quality Studies, University of Bologna, 47921 Rimini, Italy; mario.mauro4@unibo.it; 3Department of Basic Medical Sciences, Neuroscience and Sense Organs, University of Study of Bari, 70124 Bari, Italy; francesco.fischetti@uniba.it (F.F.); gianpiero.greco@uniba.it (G.G.); 4Bologna F.C. 1909 Technical Center, 40128 Bologna, Italy; lucaben759@gmail.com (L.B.); paolo.aiello1974@gmail.com (P.A.); 5Isokinetic Research Center, 40132 Bologna, Italy; g.nanni@isokinetic.com (G.N.); dimiceli.r@gmail.com (R.D.M.); 6Russi S. U., 48026 Ravenna, Italy; davide.gallamini3@studio.unibo.it

**Keywords:** soccer young players, body components, BIVA, physical abilities

## Abstract

**Simple Summary:**

In recent years, soccer teams require greater physical and technical-tactical capabilities from not to younger players, especially in elite team. Although dribble and kicking skills, strength, endurance, speed, and agility abilities are the most relevant features, it is not clear whether anthropometric and body composition aspects could be considered appropriate talent characteristics in soccer players. In addition, it rests unclear which are the principal differences, when they appear, and what metrics discriminate between elite and non-elite teams. The main aim of this study is to compare anthropometric, body composition and physical performance between and within four juvenile categories of two (elite and non-elite) soccer teams and investigates factors that better discriminate among two teams. Despite the physical performance results as the most relevant factor in discriminating among the two soccer societies, the elite players report better anthropometric and body characteristics, especially in the youngest categories.

**Abstract:**

Soccer is a multifactorial sport, in which players are expected to possess well developed physical, psychological, technical, and tactical skills. Thus, the anthropometric and fitness measures play a determinant role and could vary according to the competitive level. Therefore, the present study aimed to verify differences in body composition and physical performance between two soccer team. 162 young soccer players (from the Under 12 to Under 15 age categories; age: 13.01 ± 1.15 years) of different competitive levels (elite—*n* = 98 and non-elite—*n* = 64) were recruited. Anthropometric characteristics (height, weight, lengths, widths, circumferences, and skinfold thicknesses (SK)), bioelectrical impedance, physical performance test as countermovement jump (CMJ), 15 m straight-line sprints, Yo-Yo Intermittent Recovery Test Level 1 (Yo-Yo), and 20 + 20 m repeated-sprint ability (RSA)) were carried out. In addition, Body mass index (BMI), body composition parameters (percentage of fat mass (%F), Fat mass (FM, kg), and Fat-free mass (FFM, kg)) and the areas of the upper arm, calf and thigh were calculated, and the Bioelectric Impedance Vector Analysis (BIVA) procedures were applied. In addition, a linear discriminant analysis was assessed to determine which factors better discriminate between an elite and non-elite football team. Many differences were observed in body composition between and within each football team’s category, especially in triceps SK (*p* < 0.05), %F (*p* < 0.05), and all performance tests (*p* < 0.01). The canonical correlation was 0.717 (*F*_(7,128)_ = 19.37, *p* < 0.0001), and the coefficients that better discriminated between two teams were 15 m sprint (−2.39), RSA (1−26), suprailiac SK (−0.5) and CMJ (−0.45). Elite soccer team players present a better body composition and greater physical efficiency. In addition, BIVA outcome could be a relevant selection criterion to scout among younger soccer players.

## 1. Introduction

Soccer is practiced all over the world and has been part of Olympic competitions since 1900. It has been classified as an intermittent sport that requests many types of physical tasks as running and sprinting at various intensity, sudden accelerations, decelerations and direction changes, and several coordinative and technic-tactical skills [1,2,3].

Thus, being a multifactorial sport, players are expected to possess well developed physical, psychological, technical, and tactical skills. The talent scouting in soccer, at the juvenile level, is usually carried out early, with the principal aim to further develop their skills and competences [4]. Entering into high-level teams is an important milestone for the development of promising players since recruited players benefit from exposure to élite level coaching, sports science and medical support, training equipment and facilities, and competition [4,5,6].

The assessment of the differences between athletes of different competitive levels can provide a better understanding of the specific requirements of elite soccer players, and a valuable insight into what is truly necessary for competitive success in that sport [1]. Particularly, anthropometric measures and body composition, and both physiological and physical capabilities, including cardiorespiratory endurance, muscular strength, muscular endurance, and flexibility, are generally assessed through testing of the soccer players [1,7,8,9]. These measures can complement each other, and their combination may provide soccer coaches and athletic trainers a better understanding of those characteristics required for successful participation at the elite level. Among several capabilities, a key role seems to be played by the lower limb power, and several physical tests as repeated sprint abilities (RSA) and shuttles, Yo-Yo distance, short distance sprints, and counter-movement jump (CMJ) were assessed to evaluate soccer player physical performance [1,8]. Despite endurance abilities are needed to perform repeated movements as jumps and travel long distances up to 12 km, the ability to perform quicker sprints and higher jumps than an opponent is crucial in determining the results of duels within a match [10,11,12]. Due to these sport features and the unpredictable in activity/recovery ratio, soccer can be considered an acyclic sport with several demands [10,11]. 

Body composition is closely related to the players’ ability of achieving maximum performance in several performance tests soccer related [7]. High level of fat acts as undesirable weight in motor actions, in which the body mass must be continuously lifted against gravity and may substantially decrease the player’s performance [13]. Body fat determines the amount of bio-mechanical inertia that a soccer player must overcome when accelerating and changing direction, thus there is an incompatibility between high fat percentage and competitive excellence [13]. Low fat percentage are related to quicker sprinting, acceleration, change of direction times and are also appropriate for jumping performance [14]. In addition, some researchers found that percentage of body fat could distinguish higher from low level soccer player [15].Body characteristics and physical abilities are an important element in player scouting at the youth level [16,17,18,19,20,21]. The variability in these hallmarks are evident in the players of the distinct competitive levels and previous studies on soccer players have disclosed significant differences in anthropometric and fitness measures between playing levels [16,22,23,24,25,26]. However, the physical demands of elite senior football players have increased rapidly in recent years, and this could affect recruiters and coaches to put greater emphasis on physical fitness from an early age [19]. Thus, it is important to have updated information on the characteristics which most influence the soccer performance and understand which variables could be specific traits to reveal future high-level players. 

Therefore, this study aimed to value the differences in body characteristics and some physical abilities among the players of two Italian youth teams of different competitive levels, one elite and one non-elite and understand the main factors that differentiate them. These knowledges could have practical implication on selection and identification of attributes which should influence the elite team scouting criteria. 

## 2. Materials and Methods

### 2.1. Participants and Design

This is a cross-sectional study design assessed in December 2021. At the beginning, a sample of 191 male attending soccer adolescent players (age: 13.01 ± 1.15 years) was enrolled. Of these, 162 (from the Under 12 to Under 15 age categories) completed all the evaluations and then were analysed. 98 attending soccer players belonged to the elite team Bologna Football Club 1909^®^, whereas 64 were registered in the non-elite team Russi Sport Union 1925^®^. All the adolescents and their parents were informed and volunteered decide to participate in the study. No randomization was adopted. Participants were allocated in one of the four categories according to their age at the day of the tests.

No diet information was gathered.

The players of the élite group trained for 6 h a week (four workouts of 1.5 h each), while the players of the non-élite team trained for 4.5 h a week (three workouts of 1.5 h each).

Written informed consent was provided by the parents before the study began. The study was approved by the Bioethics Committee of the University of Bologna (Approval code: 25027).

### 2.2. Anthropometry

Anthropometric characteristics (height, weight, lengths, widths, circumferences, and skinfold thicknesses) were collected by a trained operator according to standardised procedures [27]. Height and sitting height were measured to the nearest 0.1 cm using a stadiometer (GPM, Zurich, Switzerland), and leg length was derived by the subtraction of sitting height from height. Body weight was measured to the nearest 0.1 kg (light indoor clothing, without shoes) using a calibrated analogue scale. Circumferences (relaxed and contracted upper arm, thigh, and calf) were measured to the nearest 0.1 cm with a non-stretchable tape and widths (humerus and femur) to the nearest 0.1 cm with a sliding caliper, both at left side of the body. The upper arm circumference was taken at the mid-point between the shoulder acromion and the olecranon process point, with the participant’s elbow relaxed along the body side (relaxed evaluation) or to be flexed 90° with palm facing upward (contracted evaluation); the thigh circumference was taken at the mid-point between the inguinal fold and the superior rotula point, with participant in standing position (thigh muscles relaxed); the calf circumference was taken at the bulkiest calf point, with participant in standing position (calf muscles relaxed); the humerus and femoral widths were taken, respectively, between the own lateral and medial epicondyles, with participants elbow and knee flexed 90°. Skinfold thicknesses (biceps, triceps, subscapular, supraspinale, sovrailiac, thigh, and calf) were measured to the nearest 1 mm using a Lange skinfold caliper at the left side of the body (Beta Technology Inc., Houston, TX, USA) at the following sites: triceps and biceps, vertically at the mid-point between the acromion process and the olecranon process, respectively, at the posterior and anterior upper arm face; subscapular, at an angle of 45” to the lateral side of the bod, about 20 mm below the tip of the scapula; sovrailiac, about 20 mm above the iliac crest (in the axillary line); thigh, vertically at the mid-point between the inguinal fold and the superior rotula point; calf, vertically at the bulkiest calf point both medially and laterally. 

Each anthropometric measurement was performed three times and the mean value was used. 

Body mass index was computed as weight/stature squared. Body composition parameters (percentage of fat mass), Fat mass, and Fat free mass were calculated using the skinfold equations developed by Slaughter and colleagues [28]. The total area of the upper arm, calf, and thigh, the muscle area of the upper arm, calf, and thigh, and the fat area of the upper arm, calf, and thigh were calculated according to Frisancho [29]. In addition, arm fat index, calf fat index, and thigh fat index were derived.

### 2.3. Physical Performance Tests

The performance tests were implemented at the University sports center, outdoors on a grassy surface to use the same field condition of a soccer game. Measures included Yo-Yo, Countermovement Jump Test, 15 m straight-line sprints. All the tests were preceded by a supervised and standardized warm-up consisting of 10 min of jogging, 5 min of athletic drills including jumping jack, lateral skip, high knee walk and backwards run, and 10 min of dynamic stretching of the lower limbs. Sufficient recovery time of 3 min was allowed between each performance trial. A photoelectric cell timing system (Fusion Sport Smart Speed Timing Gates, Brisbane, Australia) was used to measure the run tests (Yo-Yo, 20 m sprints), while the CMJs were measured by two photocells that estimate the distance from the field through the jump duration (OptoJump^®^, Microgate, 11 Miller Road, 10541 Mahopac, New York, NY, USA).

Yo-Yo consisted of repeated 20 m runs back and forth between the starting, turning, and finish lines at a progressively increased speed, which is controlled by audio beeps from a tape recorder. When the participants failed twice to reach the finish line in time, the distance covered was recorded as the test result. This test consists in 4 running bouts at 10–13 km·h^−1^ and another 7 runs at 13.5–14 km·h^−1^, and then continues with stepwise 0.5  km·h^−1^ speed increments after every 8 running bouts (i.e., after 760, 1080, 1400, 1720 m, etc.) until exhaustion [30]. One trial was assessed for each player.

To test the CMJ each participant was instructed to start from an upright position, making a rapid downward movement to a knee angle of 90° and simultaneously beginning to push-off [31]. Feet position coincided with the fitted acromion vertical line, with an extra-rotation at most of 15°. The hands were maintained on the waist for the entire trial. One minute of rest was allowed between the two attempts and the higher value was gathered. 

The determination of 15 m sprint times was performed on a football field and all participants wore training clothing and soccer boots, as a previous study [32]. Players were positioned behind the start line (0.5 m) and were instructed to perform the sprint with maximal effort, after a sound start signal. Two trained coaches recorded the time to complete 15 m. Each athlete performed two attempts and the mean result was gathered.

RSA test consisting of six  40 m (20 +  20 m sprints with 180° turns) shuttle sprints separated by 20 s of passive recovery was assessed as described by Rampinini et al. [33]. The athletes started from a line, sprinted for 20 m, touched a line with a foot and came back to the starting line as fast as possible. After 20s of passive recovery, the soccer players started again. Sprinting times were recorded with photoelectric cells (Fusion Sport Smart Speed Timing Gates, Brisbane, Australia). One trial was assessed for each player.

The best time (BT) in a single trial and the mean time (MT) were measured. The percentage of sprint decrement (%Sdec) was calculated as follows: 100 − (MT/BT × 100).

### 2.4. Bioelectric Impedance Vector Analysis (BIVA)

The impedance measurements were performed with bioimpedance analysis (BIA 101 Anniversary, Akern, Florence, Italy) using an electric current at a frequency of 50 kHz. Measurements were made using four electrical conductors; the subjects were in the supine position with lower limbs angle of 45° compared to the median line of the body and the upper limbs angle of 30° from the trunk. After cleansing the skin with alcohol, two Ag/AgCl low impedance electrodes (Biatrodes Akern Srl, Florence, Italy) were placed on the back of the right hand and two electrodes were placed on the corresponding foot [34]. To avoid disturbances in fluid distribution, athletes were instructed to abstain from foods and liquids for ≥4 h before the test. Athletes consumed a normal breakfast at 07:00 and the measurements were taken at 11:00. Vector length (VL) was calculated as (adjusted R^2^ + adjusted Xc^2^) 0.5 and PA as the arctangent of Xc/R × 180°/π. BIVA was carried out using the classic methods, e.g., normalizing R (Ω) and Xc (Ω) for height in meters [35]. Elite male soccer players bioelectrical specific values [36] were used as a reference to build the 50%, 75%, and 95% tolerance ellipses on the R–Xc graph

### 2.5. Statistical Analysis

Descriptive statistics (mean ± standard deviation, SD) were calculated for each variable. Variable normality was verified with the Shapiro–Wilk test. When a variable reported a *p*-value (*p*) < 0.05, a check for curve distribution skewness was assessed. Due to common right skewed function curve, in all skinfold thickness measurements a logarithm transformation was applied to meet the bell-shape distribution. 

The student’ *t*-test was performed on all anthropometric characteristics and physical performance trials to test the differences between the two teams for each category (U12, U13, U14, U15), and within each team for two categories (U13 and U15); the test value (*t*) and *p* were reported. When measurement percentage was compared, the *Z* test of proportion was used. 

In order to describe the BIVA results, each team category was plotted in the tolerance ellipses (50%, 75%, and 95%) and 10- to 11, or 12, or 13, or 14- to 15-year-old, healthy male Italian reference population. Compared to our sample, these populations represent the closest references in terms of age [37]. Then, the BIVA confidence of each category mean was calculated to compare distances among and between two teams. A two-sample Hotelling’s *T*^2^, *F*, *p* and Mahalanobis distances (*D*) were reported. Furthermore, we examined the differences between every group and the Serie A elite players [36]. 

In order to select which variable could better discriminate between the two football teams, a Linear Discriminant Analysis (LDA) through the stepwise procedure was performed. Both Fisher’s and Mahalanobi’s approaches were used [38,39]. The leave-one out average posterior probabilities classification was assessed to see how many observations were correctly classified in each group. The MANOVA statistic was performed and the values of Wilk’s lambda, Pillai’s trace, and Lawley-Hotelling trace were reported. Due to a high Snedecor-Fisher (*F*) value and significant *P*, the univariate ANOVA was computed, and the goodness of fit (*R*^2^ and adj. *R*^2^) *F* and *p* were reported for all variables included in the regression model. Since we just had two groups (Bologna and Russi), only one discriminant function was produced; the canonical correlation value, eigenvalue, Likelihood Ratio (LRR), *F* and *p* were reported. To obtain a projection of the data that gave us maximal separation between the two groups, each standardized (using the pooled within-group covariance matrix) coefficient of the discriminant function was reported. These coefficients are appropriate for interpreting the importance and relationship of the discriminating variables within the discriminant functions, where higher absolute value indicates an important role of the related variable in the discrimination function. In addition, the squared Mahalanobis distance was calculated and the *D*^2^ and *p* were reported.

A *p*-value (*p*) < 0.05 was considered significant. To avoid the type one error inflation, a priori test was assessed both in Fisher’ (Hotelling’s *T*^2^*)* and Student’ (*t*) family tests by G*Power 3.1.9.7 software (edition for Windows) [40,41]. The input parameters for the F test were effect size = 0.5, α = 0.05, 1 − β = 0.80, allocation ratio = 0.65, while the sample size requested was of 100 participants in group one and 65 in group two. The input parameters for the two-tailed *t* test (difference between two independent means) were the same, but the sample size requested was of 81 participants in group one and 53 in group two.

In within team analysis where more than one group comparison was performed, a Bonferroni correction was applied to avoid one type-error inflation (α/*m*, where *m* = number of comparisons). BIVA software [42] were used for all statistical calculations BIVA related. It allows to plot individuals in the tolerance ellipses (50%, 75%, and 95%) of a reference population. These ellipses are obtained from the literature using the population size, mean, and SD of both R/H and Xc/H, with their linear correlation coefficient. Furthermore, BIVA software allows the calculation of the two-sample Hotelling’s *T*^2^ test and the Mahalanobis *D*, by means of the same descriptive variables. The Other statistical analysis was performed with STATA^®^ software for Windows 10, version 17 (Publisher: StataCorp. 2021. Stata Statistical Software: Release 17. College Station, TX, USA, StataCorp LP).

## 3. Results

Table 1 shows mean and standard deviation of each variable for all categories of each team and the statistical differences between them. Among the anthropometric variables, elite soccer players were generally taller than non-elite peers, with significant differences in U12 and U14.Elite soccer players U12 presented significant lower values than non-elite in thigh circumference and femoral dimeter, in biceps, triceps and medial and lateral calf skinfolds and in calf fat area and calf fat index.

The U13 represented the category which presented the most marked differences between the two groups, since BMI, circumferences (with the exception of calf), humeral diameter, skinfold thicknesses, fat mass and the majority of the limb areas significantly differed. In addition, Bologna U13 showed significant higher PA values than Russi U13. As regards skinfold thicknesses, significant differences were observed between the two groups also in U14, with the exception of medial calf skinfold. In U15 the differences between the two groups were very small, regarding, in addition to the triceps skinfold, only suprailiac and medial calf skinfolds. 

No significant differences result within each category in age, weight, calf circumference, and calf muscle area. 

As regards body composition parameters fat mass showed significant difference between elite and non-elite U13 and U14, while fat free mass did not report relevant differences. If percentage of fat mass is considered, elite player of all the categories presented significant lower values than non-elite peers.

Phase angle significantly differed only in U13, while R/H and Xc/H only in U12.

As regards the physical performance/motor tests/fitness capacities/motor abilities, all the considered variables (CMJ, 15 m sprint, RSA) showed significant differences between the two groups in each age category. In addition, the YO-YO test reports significant differences among U14 (*t* = 10.21, *p* < 0.001) and U15 (*t* = 3.87, *p* < 0.001) categories. 

Table 2 shows the mean differences among the F.C. Bologna U13 and U15, and among U.S. Russi U13 and U15 categories. Generally, younger categories presented higher values of skinfold measures when compared with elder soccer players. Both U15 categories reported higher value of calf muscle area and fat free mass than younger players, and the lowest values of fat mass percentage and calf fat index. In addition, the elder categories showed better physical performance outcomes in counter movement jump, 15 m sprint and repeated sprint ability tests. 

### 3.1. Bioelectrical Impedance Vector Analysis

Figure 1 shows the BIVA confidence outcomes among and between the categories of the Bologna and Russi Football Teams. Picture A displays the differences among each group (U12 = Group 1, U13 = Group 3, U14 = Group 5, U15 = Group 7). Only one comparison (U14 vs U15) did not show significant differences. A trend with increasing age was observed for the increase of cell mass and tissue hydration. Differently, the comparisons among Russi categories (picture B) reported significant differences only between the Group 2 (U12) and the other groups, which indicates that relevant changes are visible up to 13 years old.

Figure 1C shows the difference between all Bologna and Russi team players: the elite football team (Bologna) reported better values in term of cellularity status (and hydration. In addition, Figure 1D displayed the comparisons between Bologna U12 and Russi U13, U14, U15, and showed only a significant difference with Russi U15.

Figure 2A shows BIVA confidence distances between adult Serie A football players (data from Micheli, [31]) and Bologna, and Russi. Although large significant differences were observed both for Bologna and Russi teams, the distance was lower for the elite players team. Figure 2B displays impedance vectors of all categories of Bologna and Russi teams plotted on the 50%, 75%, and 95% tolerance ellipses of adult football players. It is evident that younger categories of the elite football team were less distant than those of Russi team. In addition, the differences among the elder and younger categories were more pronounced in elite team where body composition of U14 and U15 players were very close to the reference (filled square and diamond).

### 3.2. Linear Discriminant Analysis (LDA)

In order to report the outcomes of the LDA, the Leave-one-out average-posterior-probabilities classification reports that the total average posterior probability for Bologna Team is 88.3% and for Russi Team is 82.9%. The MANOVA test show significant results for each test (*n* = 136, *F*_(7,128)_ = 19.37, *p* < 0.001). The univariate ANOVA test shows all variables, which present significant outcomes, where the motor tests exhibit higher values of the goodness of fit (15 m Sprint: *adj. R*^2^ = 0.401; RSA: *adj. R*^2^ = 0.292; CMJ: *adj. R*^2^ = 0.144), followed by biceps (*adj. R*^2^ =0.111), suprailiac (*adj. R*^2^ = 0.088), medial calf (*adj. R*^2^ = 0.088) and triceps (*adj. R*^2^ = 0.082) skinfold thicknesses. In addition, the canonical LDA function and the coefficients standardized using the pooled within-group covariance matrix was calculated. The canonical correlation equals to 0.717 (*LLR* = 0.486; *F*_(7,128)_ = 19.37, *p* < 0.001) and the 15 m Sprint reports the highest coefficient absolute value (−2.39) that indicates the most contributory factor in discriminating between the teams, followed by the RSA test (1.26), suprailiac skinfold (−0.5) and CMJ test (−0.45) as reported by the ANOVA outcomes. Finally, we calculate the squared Mahalanobis distance which is equal to 4.49 (*F*_(7,128)_ = 19.37, *p* < 0.0001). All these contents are presented as Appendix A.

## 4. Discussion

The aims of the present study were to value the differences in anthropometric characteristics and physical performance among the attending soccer players of two Italian youth teams (from U12 up to U15) of different competitive levels, one elite and one non-elite, in order to understand which traits could differentiate between elite and non-elite soccer teams and in which juvenile categories they appear.

As regards the anthropometric parameters, the selected players were generally taller compared to their non-selected counterparts, even is significant differences were observed only in U12 and U14 categories. This is in line with previous research showing that adult players attaining higher levels of play were, on average, substantially differentiated from amateur players in height as well as body mass [23,43].

Of particular note is that U13 elite soccer players presented the most marked differences in comparison with low level peers, since BMI, circumferences (with the exception of calf), skinfold thicknesses, fat mass, fat mass percentage and the majority of the limb areas significantly differed from their low-level counterparts. In addition, Bologna under 13 showed significant higher phase angle values than Russi U13 categories, indicating better cell integrity and functionality. This suggests that this category, probably because of the differences linked to the particular period of growth, it is the one that deserves special attention.

Among the anthropometric characteristics, the results of this study showed that skinfolds are the parameters that differ most between the two groups. This confirms the importance of monitoring body fat, as appropriate fat levels enable players to move more effectively during training and games [44,45]. In particular, the triceps skinfold showed significant differences between competitive levels in each category. Apart from U13, significant differences were observed between the two groups also in U14 players, except for medial calf skinfold.

In U15 category the differences between the two groups were very small, regarding, in addition to the triceps skinfold, only suprailiac and medial calf skinfolds. This could suggest that the differences between the players of the two teams become more attenuated with aging. However, this study did not consider the maturity assessment and many factors could affect our results.

Considering the difference from elite and non-elite groups, from the results emerged that all the physical performance variables showed significant differences between the two groups in each age category. This confirm that high-intensity activities are fundamental aspects for performance in soccer [43,46]. The elite players were capable of higher acceleration over 15 m than non-elite players. This is in accordance with previous studies which showed that elite players tended to present better sprint performances and change of direction than non-elite ones [24,43,47]. The differences in sprint time could be connected to the fact that elite players predominantly perform their high-speed runs over short distances during the match [48]. The better results in repeated sprint ability test showed by the elite player in comparison with non-elite is in accordance with previous studies carried out on players of different age groups [33,49,50]. In the current study, U14 and U15 elite players presented longer distances covered during Yo-Yo test compared to non-elite ones.

In relation to the age, both elite and non-elite players showed a growing trend for some anthropometric measurements and all the physical tests: the under-15 division registered higher physical performance and better body composition values than the other categories, especially in fat free mass. In addition, fat mass percentage decreased with age in both groups. This trend is in agreement with the observation of Slimani and colleagues, who reported that, as compared to the older groups (U17, U19, and Pro2), the U15 players have a significantly higher % of body fat [2]. Many authors found a negative high correlation between the body fat percentage and the physical performance in elite players, which could indicate that the body composition impacts performance outcomes [45,51,52]. Elite players of all the categories considered in the present study showed a significant lower %F than non-elite players. Previous studies on soccer players have provided significant differences in body fat percentage between soccer player of different levels [2]. The overall of % of body fat mean values reported in the scientific literature vary between 9.9 and 11.9% for male elite, and between 12.4 and 16.5% for amateur senior soccer players [2]. As regards %F, elite and non-elite U13 and U14 player fall within their respective ranges.

In terms of phase angle, elite soccer players showed generally higher values than non-elite, even if, as already reported, the differences were only significant for U13. Apart from U12, the mean phase angle values found in elite players of the present study were comparable than those reported in prior studies with age-matched male athletes (U13 to U17: range = 6.2–7.0°) [9,53,54,55], and higher than those reported by Martins and collaborators on U13 and U15 Brazilian professional soccer [56]. It has been shown that PA is an objective indicator of cellular health, with higher values reflecting better cellularity, cell membrane integrity, and cell function, while lower phase angle values can indicate decreased cell integrity [56]. Considering that phase angle is a measure derived from resistance and reactance, any alteration in cellular membrane integrity (Xc), body fluid (R), or a combination of both, results in changes in phase angle.

With regard to bioimpedance vector analysis, the present study showed significant differences in confidence ellipses among and between the categories of the elite and non-elite soccer teams. Among the age categories, a trend was observed for the increase of cell mass and tissue hydration, especially in elite team where significant distances were found in all comparisons (except between U14 and U15 groups). Differently, the comparisons among non-elite team categories reported significant differences only between U12 and the other groups, which indicates less differentiation between age categories and a greater homogeneity. As regards the differences between the categories, the elite soccer team reported better values in term of cellularity status and hydration than non-elite team. In addition, the comparisons between elite U12 and non-elite U13, U14, U15, carried out to understand whether BIVA outcome could be a relevant selection criteria and parameter to scout among adolescent football players, showed only a significant difference with non-elite U15, which indicates that elite adolescent players exhibit similar cellular composition when compared with elder players of non-elite team. BIVA confidence distances between adult Serie A football players (data from Micheli, [31]) and elite and non-elite groups showed that although significant differences were observed both for elite and non-elite teams, the distance was lower for the elite players team. The younger categories of the elite soccer team were less distant from the reference ellipses than those of non-elite team, which may indicate that elite team has strict selection criteria or begins the scouting process earlier. In addition, the differences among the elder and younger categories were more pronounced in elite team where body composition of U14 and U15 players were very close to the reference.

The second aim was to identify the minimum set of predictors that best discriminate the elite and non-elite groups, in order to provide important and useful information that may help coaches to improve the development and selection of young players, as well as to increase success opportunities in their training sessions and competitions. The variables that best discriminated the two groups were Sprint 15 m, repeated sprint ability, and countermovement jump, in terms of physical performance, followed by suprailiac, triceps, medial calf and biceps skinfold thicknesses. The Sprint 15 m reports the highest absolute value, which indicates the most contributory factor in discriminating between the two groups, followed by the repeated sprint ability test, suprailiac skinfold and countermovement jump test. Therefore, coaches and practitioners should consider these characteristics over the talent identification and development process. It is important to note that, apart from motor tests, skinfold thicknesses may well guide training programs, having the potential to associate with competitive level and match performance. Nughes and colleagues in their study on anthropometric and functional profile of selected vs. non-selected 13-to-17-year-old soccer players found that that dribbling skills, 15-m sprint time, and height best discriminate U17 players by competitive level [43]. Contrarily to the results of the present study, anthropometric characteristics and functional abilities could not discriminate across competitive standards between younger male (U15), but only U17 soccer players.

The results of the present study could have practical implication on the trainability or not of the identified components and on the strategies to be adopted. However, it must be taken into consideration that the selection of soccer players is a strongly debated issue, since scouts do assess and advise on selection of players at younger ages. According to Bergkamp scouts are aware of the idea that early indicators of later performance are often lacking or hard to predict, given the difficulty of predicting future performance directly [57]. Scouting in the younger age cohorts could be more affected by the finding of the best current player, rather than finding the best player for the future [58]. This approach seems to rely on the assumption that the best current young players are also those that have the highest potential for excellence in the future. In any case, even on the youngest, this study reveals useful suggestions on the most informative parameters for selective purposes.

The main limitation of this study is the lack of an assessment of biological maturation. This study investigated 12-15-year-old players who were homogeneous in terms of chronological age, and the of growth and maturation process could have interfered with their anthropometric characteristics and physical test measures. Furthermore, we did not consider playing position, but it is to consider that the physical demands that characterize the specific positional roles require soccer players to adapt to meet them, influencing their characteristics. Finally, we did not compare groups with the same sample dimension and no randomized group allocation was applied during the sampling process.

## 5. Conclusions

Elite soccer team players present better anthropometric characteristic and higher physical performance level than non-elite players. Although many untreated factors should influence the physical growth, the age plays a key role in increasing the body composition and capabilities in both elite and non-elite soccer players. Despite this, elite youngest players reveal BIVA outcomes closer to older groups and it may be a relevant selection criterion to scout among adolescent soccer players. Although the physical performances are the most discriminant factors between elite and non-elite teams, the body composition deserves a greater focus juvenile soccer research.

## Figures and Tables

**Figure 1 biology-11-00823-f001:**
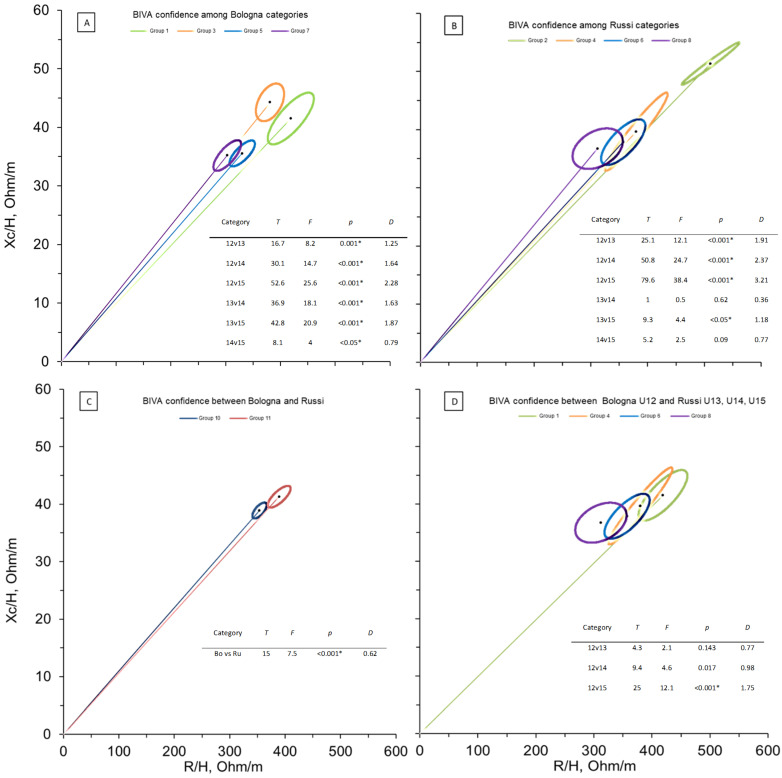
BIVA confidence among and between Bologna and Russi categories: (**A**) among Bologna categories (Group 1 = U12, Group 3 = U13, Group 5 = U14, Group 7 = U15); (**B**) among Russi categories (Group 2 = U12, Group 4 = U13, Group 6 = U14, Group 8 = U15); (**C**) between Bologna and Russi mean; (**D**) between Bologna U12 (Group 1) and Russi U13 (Group 4), U14 (Group 6), U15 (Group 8). Note: *T*, Hotelling *T*^2^; *F*, Snedecor-Fisher test; *p*, *p*-value; *D*, Mahalanobis distance; *, statistically significant.

**Figure 2 biology-11-00823-f002:**
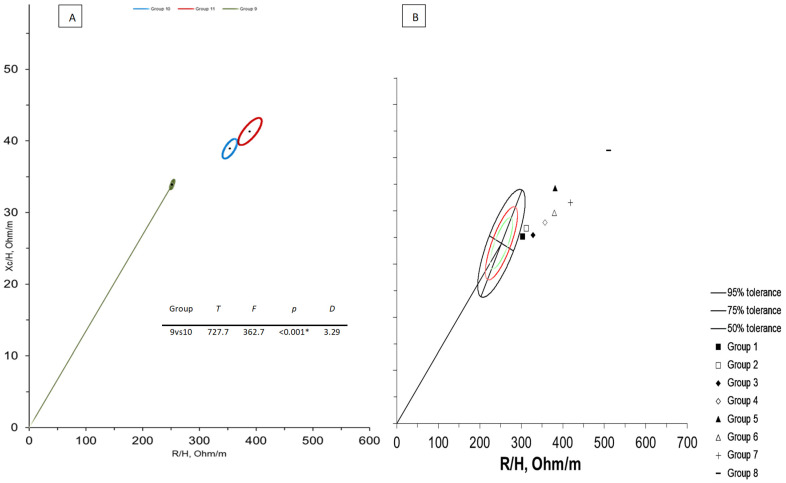
(**A**) BIVA confidence between adult Serie A football players (Group 9, green vector), Bologna (Group 10, blue vector) and Russi (Group 11, red vector) adolescent players. (**B**) BIVA tolerance between adult Serie A football players (Micheli vector, [31]) and Bologna U15 (Group 1), Russi U15 (Group 2), Bologna U14 (Group 3), Russi U14 (Group 4), Bologna U13 (Group 5), Russi U13 (Group 6), Bologna U12 (Group 7), and Russi U12 (Group 8). Note: *T*, Hotelling *T*^2^; *F*, Snedecor-Fisher test; *p*, *p*-value; *D*, Mahalanobis distance; *, statistically significant.

**Table 1 biology-11-00823-t001:** Variable statistics of Bologna and Russi Calcio for each category.

Variable	Bo U12 (18)	Ru U12 (16)	Bo U13 (27)	Ru U13 (12)	Bo U14 (30)	Ru U14 (21)	Bo U15 (23)	Ru U15 (15)	∆ U12		∆ U13		∆ U14		∆ U15	
	Mean (±SD)	Mean (±SD)	Mean (±SD)	Mean (±SD)	Mean (±SD)	Mean (±SD)	Mean (±SD)	Mean (±SD)	*t*	*p*	*t*	*p*	*t*	*p*	*t*	*p*
Age	11.38 (0.36)	11.37 (0.28)	12.35 (0.25)	12.37 (0.33)	13.44 (0.24)	13.49 (0.26)	14.36 (0.31)	14.43 (0.34)	0.10	0.918	−0.189	0.849	−0.781	0.443	−0.612	0.541
Weight (Kg)	41.30 (7.10)	39.59 (8.41)	43.89 (6.15)	48.42 (8.35)	52.98 (8.04)	52.43 (11.08)	62.79 (9.04)	58.63 (13.19)	0.634	0.530	−1.896	0.066	0.213	0.841	1.154	0.261
Height (cm)	153.00 (8.20)	142.50 (4.77)	154.96 (7.63)	153.90 (9.47)	165.91 (8.28)	160.93 (8.31)	173.30 (8.99)	169.77 (7.92)	4.423	<0.001 *	0.372	0.711	2.112	0.039 *	1.214	0.223
BMI (kg/m^2^)	17.54 (1.75)	19.42 (3.70)	18.21 (1.53)	20.36 (2.73)	19.14 (1.52)	20.11 (3.31)	20.88 (2.55)	20.21 (3.38)	−1.932	0.062	−3.149	0.003 *	−1.412	0.163	0.700	0.491
Rel. arm circum. (cm)	20.88 (2.00)	22.04 (3.23)	21.20 (1.58)	23.76 (2.27)	22.94 (1.94)	24.09 (2.88)	24.32 (1.73)	24.54 (3.27)	−1.274	0.212	−4.060	<0.001 *	−1.701	0.104	−0.266	0.789
Cont. arm circum. (cm)	22.24 (2.00)	23.36 (3.25)	23.24 (1.76)	25.38 (2.04)	25.12 (2.13)	26.10 (3.42)	26.66 (3.52)	26.41 (3.39)	−1.224	0.229	−3.331	0.002 *	−1.251	0.223	0.212	0.831
Calf circum. (cm)	30.57 (2.51)	31.13 (3.29)	30.94 (4.06)	33.25 (2.28)	33.99 (4.48)	33.69 (2.93)	34.55 (2.38)	34.93 (3.28)	−0.560	0.579	−1.791	0.074	0.267	0.789	−0.412	0.691
Thigh circum. (cm)	40.86 (3.49)	44.11 (5.35)	40.84 (3.85)	46.63 (4.00)	45.16 (3.62)	47.60 (5.78)	46.28 (3.88)	48.33 (5.77)	−2.122	0.041 *	−4.286	<0.001 *	−1.861	0.069	−1.319	0.199
Humeral diameter (cm)	5.88 (0.39)	5.74 (0.42)	6.11 (0.33)	6.07 (0.50)	6.41 (0.33)	6.43 (0.42)	6.71 (0.30)	6.66 (0.41)	0.965	0.342	−4.287	<0.001 *	−0.241	0.812	0.463	0.651
Femoral diameter (cm)	8.60 (0.44)	8.94 (0.57)	8.61 (0.51)	8.97 (1.02)	9.27 (0.54)	9.35 (0.55)	9.42 (0.46)	9.73 (0.58)	−1.976	0.054 *	0.271	0.789	−0.530	0.603	−1.841	0.071
Biceps SK (mm)	1.67 (0.38)	1.99 (0.40)	1.50 (0.30)	1.92 (0.47)	1.48 (0.28)	1.72 (0.44)	1.28 (0.20)	1.56 (0.37)	−2.351	0.024 *	−2.878	0.010 *	−2.231	0.034 *	−2.711	0.010 *
Triceps SK (mm) ^#^	2.13 (0.33)	2.36 (0.30)	2.09 (0.21)	2.36 (0.35)	1.90 (0.30)	2.22 (0.35)	1.85 (0.28)	2.01 (0.38)	−2.163	0.039 *	−2.502	<0.017 *	−3.374	0.001	−1.371	0.183
Subscapular SK (mm) ^#^	1.75 (0.27)	1.88 (0.48)	1.67 (0.17)	2.12 (0.34)	1.80 (0.16)	2.02 (0.4)	1.88 (0.21)	1.88 (0.32)	−0.978	0.345	−4.389	<0.001 *	−2.419	0.021 *	−0.011	0.999
Supraspinal SK (mm) ^#^	1.79 (0.37)	1.96 (0.54)	1.54 (0.25)	2.05 (0.47)	1.57 (0.22)	1.94 (0.48)	1.67 (0.21)	1.84 (0.41)	−1.113	0.282	−3.491	0.002 *	−3.34	<0.01 *	−1.501	0.153
Suprailiac SK (mm) ^#^	2.09 (0.38)	2.29 (0.48)	1.96 (0.27)	2.41 (0.35)	2.04 (0.25)	2.30 (0.44)	2.06 (0.25)	2.25 (0.33)	−1.331	0.202	−3.967	0.001 *	−2.446	<0.020 *	−1.902	0.071
Thigh SK (mm) ^#^	2.35 (0.23)	2.50 (0.28)	2.27 (0.18)	2.46 (0.33)	2.20 (0.21)	2.42 (0.38)	2.12 (0.21)	2.28 (0.37)	−1.758	0.091	−1.959	0.067	−2.401	<0.022 *	−1.479	0.152
Medial Calf SK (mm) ^#^	1.99 (0.39)	2.31 (0.29)	1.91 (0.28)	2.23 (0.28)	1.88 (0.28)	2.04 (0.38)	1.76 (0.23)	1.96 (0.38)	−2.761	0.009 *	−3.243	0.002 *	−1.641	0.112	−1.802	0.091
Lateral Calf SK (mm) ^#^	2.08 (0.34)	2.34 (0.29)	2.00 (0.23)	2.27 (0.27)	2.00 (0.28)	2.17 (0.35)	1.92 (0.20)	2.03 (0.38)	−2.434	<0.05 *	−3.071	0.002 *	−1.839	0.072	−1.011	0.323
Total Upper area (cm^3^) ^#^	3.54 (0.20)	3.64 (0.29)	3.57 (0.15)	3.78 (0.19)	3.73 (0.17)	3.82 (0.24)	3.85 (0.14)	3.85 (0.25)	−1.113	0.282	−3.578	0.001 *	−1.511	0.142	−0.110	0.910
Upper Muscle area (cm^3^)	27.98 (6.17)	32.78 (11.24)	28.61 (4.92)	38.08 (8.76)	33.75 (6.50)	39.00 (11.26)	38.22 (6.32)	40.17 (13.67)	−1.567	0.126	−4.334	<0.001 *	−2.101	0.048 *	−0.601	0.550
Upper Fat area (cm^3^)	7.04 (0.74)	6.69 (0.56)	7.38 (0.68)	7.23 (1.04)	8.44 (0.91)	7.82 (0.98)	9.09 (0.81)	8.57 (0.87)	1.576	0.125	0.531	0.602	2.321	0.024 *	1.881	0.070
Upper Fat Index (%) ^§^	20.72 (4.04)	18.20 (4.95)	19.47 (3.48)	16.57 (4.15)	20.3 (2.34)	17.51 (3.95)	19.43 (2.05)	18.54 (4.03)	1.631	0.113	4.141	<0.001 *	3.171	0.002 *	0.901	0.374
Total Calf area (cm^3^) ^#^	4.30 (0.17)	4.34 (0.21)	4.32 (0.26)	4.47 (0.14)	4.51 (0.22)	4.50 (0.18)	4.55 (0.14)	4.57 (0.18)	−0.489	0.618	−2.412	0.023 *	0.211	0.829	−0.331	0.741
Calf Muscle area (cm^3^)	52.02 (7.78)	48.09 (8.76)	56.56 (18.29)	58.60 (7.57)	70.46 (26.86)	63.59 (10.17)	74.41 (10.34)	71.85 (8.98)	1.389	0.171	−0.371	0.713	1.119	0.271	0.792	0.443
Calf Fat area (cm^3^)	22.86 (7.87)	29.88 (10.78)	20.93 (5.92)	29.81 (8.76)	23.05 (7.38)	27.43 (11.21)	21.07 (5.23)	26.07 (12.34)	−2.184	0.041 *	−3.729	<0.001 *	−1.690	0.104	−1.732	0.093
Calf Fat index (%) ^§^	29.95 (7.78)	37.66 (7.21)	27.60 (6.51)	33.34 (6.89)	25.01 (6.33)	29.50 (8.38)	22.03 (3.92)	25.62 (7.38)	−2.983	0.005 *	−2.503	0.022 *	−2.182	0.034 *	−1.959	0.064
Total Thigh area (cm^3^)	133.81 (22.36)	157.02 (37.82)	133.96 (23.17)	174.31 (29.36)	163.36 (26.38)	182.96 (44.42)	171.66 (28.2)	188.47 (46)	−2.211	0.034 *	−4.532	<0.001 *	−1.973	0.052 *	−1.401	0.170
Thigh Muscle area (cm^3^)	120.80 (22.20)	144.45 (35.58)	120.27 (22.44)	160.63 (30.37)	147.85 (25.46)	168.88 (45.21)	155.26 (27.49)	172.99 (46.97)	−2.221	0.034 *	−4.638	<0.001 *	−2.119	0.040 *	−1.472	0.155
Thigh Fat area (cm^3^)	13.02 (1.14)	12.57 (1.40)	13.69 (1.83)	13.68 (1.99)	15.51 (1.71)	14.09 (2.88)	16.40 (1.65)	15.48 (2.18)	1.041	0.313	0.010	0.999	2.209	0.037 *	1.492	0.153
Thigh Fat index (%) ^§^	10.00 (1.87)	8.55 (2.50)	10.39 (1.43)	8.15 (2.22)	9.64 (1.22)	8.20 (2.56)	9.72 (1.22)	8.72 (2.40)	1.909	0.0644	3.789	<0.001 *	2.678	0.010 *	1.701	0.103
Fat Mass (kg) ^#^	1.72 (0.43)	1.87 (0.52)	1.74 (0.23)	2.16 (0.43)	1.77 (0.26)	2.04 (0.46)	1.76 (0.40)	1.84 (0.53)	−0.849	0.404	−3.191	<0.005 *	−2.421	0.021 *	−0.454	0.661
Fat Free Mass (kg)	35.21 (5.44)	32.32 (5.58)	38.02 (5.14)	39.02 (6.17)	46.92 (7.44)	43.86 (8.67)	56.50 (7.63)	51.39 (9.53)	1.526	0.137	−0.531	0.602	1.348	0.181	1.829	0.082
%FM ^§^	14.34 (3.94)	17.54 (5.61)	13.29 (1.98)	18.98 (5.71)	11.50 (2.73)	15.94 (5.99)	9.89 (3.11)	11.74 (4.82)	−1.913	0.054 *	−3.362	<0.001 *	−3.172	<0.001 *	−6.941	<0.001 *
Phase Angle ^#^	1.73 (0.10)	1.76 (0.05)	1.89 (0.14)	1.78 (0.06)	1.82 (0.08)	1.80 (0.13)	1.90 (0.08)	1.91 (0.29)	−1.256	0.224	3.544	0.001 *	0.421	0.681	−0.192	0.851
R/H (Ω/m)	412.09 (64.21)	511.44 (71.74)	382.83 (49.22)	383.34 (63.66)	330.14 (48.38)	359.60 (65.37)	304.38 (45.02)	312.73 (59.93)	−3.876	<0.001 *	−0.021	0.982	−1.848	0.072	−0.492	0.633
Xc/H (Ω/m)	41.73 (6.74)	51.43 (5.14)	44.42 (6.16)	40.03 (7.74)	35.55 (4.60)	38.05 (6.58)	35.36 (4.66)	36.73 (4.72)	−4.666	<0.001 *	1.879	0.072	−1.601	0.119	−0.878	0.391
YOYO test (s)	/	/	/	/	2367.40(536.90)	787.80 (461.90)	2500.00 (598.90)	1702.90 (551.30)	/	/	/	/	10.198	<0.001 *	3.871	<0.001 *
CMJ test (cm)	27.99 (2.88)	24.04 (5.13)	28.63 (3.86)	24.24 (4.78)	32.77 (2.99)	27.93 (6.27)	36.60 (6.15)	28.57 (4.53)	2.81	<0.01 *	2.941	0.005 *	3.687	<0.001 *	4.343	<0.001 *
Sprint 15 m test (s)	2.71 (0.12)	3.18 (0.19)	2.83 (0.11)	3.07 (0.19)	2.51 (0.93)	2.94 (0.23)	2.41 (0.12)	2.76 (0.13)	−8.6	<0.001 *	−4.812	<0.001 *	−9.212	<0.001 *	−8.732	<0.001 *
RSA 20 + 20 m (s)	6.34 (0.22)	7.22 (0.42)	6.57 (0.19)	6.88 (0.39)	5.84 (0.19)	6.64 (0.49)	5.69 (0.20)	6.20 (0.22)	7.27	<0.001 *	−3.042	0.004 *	2.031	0.001 *	−6.592	<0.001 *

Note: Bo, F.C. Bologna.; Ru, U.S. Russi; U12, Under 12; U13, Under 13; U14, Under 14; U15, Under 15; *t*, student’s *t*; *p*, *p*-value; circum, circumference; rel, relaxed; cont, contracted; SK, skinfold; CMJ, counter movement jump; RSA, repeated sprint ability; SD, standard deviation; * statistically significant; ∆ difference between; ^#^ logarithmic scale; ^§^ proportion analysis with the *Z*-test.

**Table 2 biology-11-00823-t002:** Mean differences among U13 and U15 categories of each team.

Variable	Bo U13 (45)	Bo U15 (53)	Ru U13 (28)	Ru U15 (36)	∆ Bologna (U13−U15)				∆ Russi (U13−U15)			
	Mean (±SD)	Mean (±SD)	Mean (±SD)	Mean (±SD)	*t*	*p*	95% CI		*t*	*p*	95% CI	
Weight (kg)	42.84 (6.58)	57.24 (9.73)	43.37 (9.35)	55.01 (12.22)	−8.421	<0.001 *	−17.79	−11.00	−4.170	<0.001 *	−17.21	−6.07
Height (cm)	154.12 (7.84)	169.12 (9.28)	147.39 (9.06)	164.61 (1.53)	−8.562	<0.001 *	−18.48	−11.52	−7.489	<0.001 *	−21.82	−12.63
BMI (kg/m^2^)	17.95 (1.64)	19.89 (2.19)	19.82 (3.29)	20.15 (3.29)	−4.921	<0.001 *	−2.73	−1.16	−0.391	0.690	−1.98	1.33
Rel. arm circum. (cm)	21.07 (1.75)	23.54 (1.96)	22.78 (2.94)	24.27 (3.00)	−6.513	<0.001 *	−3.21	−1.71	−1.994	0.050 *	−3.00	0.004
Cont. arm circum (cm).	22.84 (1.90)	25.79 (2.89)	24.22 (2.93)	26.23 (3.36)	−5.850	<0.001 *	−3.95	−1.95	−2.501	0.010 *	−3.61	−0.40
Calf circumf. (cm)	30.79 (3.50)	34.23 (3.70)	32.04 (3.05)	34.21 (3.10)	−4.699	<0.001 *	−4.89	−1.99	−2.796	0.004 *	−3.72	−0.62
Thigh circumf. (cm)	40.85 (3.67)	45.64 (3.74)	45.19 (4.90)	47.91 (5.71)	−6.291	<0.001 *	−6.29	−3.30	−2.011	0.049 *	−5.42	−0.14
Humeral diamet.(cm)	6.02 (0.37)	6.54 (0.35)	5.89 (0.48)	6.53 (0.42)	−7.198	<0.001 *	−0.67	−0.38	−5.720	<0.001 *	−0.87	−0.42
Femoral diamet. (cm)	8.60 (0.48)	9.34 (0.51))	8.95 (0.78)	9.51 (0.59)	−7.321	<0.001 *	−0.93	−0.53	−3.291	0.001 *	−0.90	−0.22
Biceps SK (mm) ^#^	1.57 (0.34)	1.4 (0.26)	1.96 (0.42)	1.65 (0.42)	2.841	0.006 *	0.05	0.30	2.878	0.007 *	0.09	0.52
Triceps SK (mm) ^#^	2.1 (0.26)	1.88 (0.29)	2.36 (0.31)	2.13 (0.37)	3.971	<0.001 *	0.11	0.34	2.567	0.010 *	0.05	0.4
Subscapular SK (mm) ^#^	1.71 (0.21)	1.84 (0.19)	1.99 (0.43)	1.96 (0.37)	−3.172	0.002 *	−0.21	−0.05	0.222	0.831	−0.18	0.23
Supraspinal SK (mm) ^#^	1.64 (0.32)	1.61 (0.22)	2 (0.50)	1.9 (0.45)	0.491	0.631	−0.09	0.14	0.812	0.424	−0.14	0.34
Suprailiac SK (mm) ^#^	2.01 (0.32)	2.05 (0.25)	2.34 (0.42)	2.28 (0.39)	−0.629	0.53	−0.15	0.08	0.611	0.542	−0.14	0.27
Thigh SK (mm) ^#^	2.30 (0.21)	2.17 (0.21)	2.49 (0.30)	2.36 (0.38)	3.111	0.003 *	0.05	0.21	1.498	0.141	−0.04	0.07
Medial Calf SK (mm) ^#^	1.94 (0.33)	1.83 (0.26)	2.27 (0.28)	2.01 (0.38)	1.851	0.072	−0.01	0.23	3.078	0.005 *	0.09	0.44
Lateral Calf SK (mm) ^#^	2.03 (0.28)	1.96 (0.25)	2.31 (0.27)	2.11 (0.37)	1.270	0.209	−0.04	0.17	2.411	0.021 *	0.03	0.37
Total Upper area (cm^3^) ^#^	3.56 (0.17)	3.78 (0.17)	3.7 (0.26)	3.83 (0.24)	−6.461	<0.001 *	−0.29	−0.15	−2.051	0.046 *	−0.26	−0.003
Upper Muscle area (cm^3^)	28.35 (5.40)	35.69 (6.74)	35.05 (10.42)	39.48 (12.15)	−5.873	<0.001 *	−9.81	−4.86	−1.541	0.130	−10.19	1.33
Upper Fat area (cm^3^)	7.25 (0.72)	8.72 (0.92)	6.92 (0.83)	8.13 (0.99)	−8.739	<0.001 *	−1.80	−1.14	−5.192	<0.001 *	−1.69	−0.75
Upper Fat Index (%) ^§^	20.74 (3.03)	19.92 (2.24)	17.5 (4.61)	17.94 (3.96)	0.101	0.920	−0.15	0.17	−0.041	0.959	−0.19	0.18
Total Calf area (cm^3^) ^#^	4.31 (0.23)	4.53 (0.19)	4.39 (0.19)	4.53 (0.18)	−5.049	<0.001 *	−0.30	−0.13	−2.781	0.011 *	−0.23	−0.04
Calf Muscle area (cm^3^)	54.75 (15.04)	72.18 (21.25)	52.59 (9.70)	67.03 (10.41)	−4.610	<0.001 *	−24.94	−9.92	−5.672	<0.001 *	−19.53	−9.35
Calf Fat area (cm^3^)	21.70 (6,75)	22.19 (6.55)	29.85 (9.79)	26.87 (11.54)	−0.361	0.723	−3.16	2.18	1.087	0.281	−2.46	8.43
Calf Fat index (%) ^§^	28.53 (7.05)	23.71 (5.57)	35.8 (7.28)	27.88 (8.10)	3.702	<0.001 *	0.20	0.30	4.111	<0.001 *	0.45	0.27
Total Thigh area (cm^3^) ^#^	4.88 (0.19)	5.1 (0.17)	5.08 (0.22)	5.19 (0.24)	−6.241	<0.001 *	−0.29	−0.15	−1.974	0.053 *	−0.23	−0.001
Thigh Muscle area (cm^3^)	120.48 (22.1)	151.07 (26.36)	151.39 (35.63)	170.59 (45.33)	−6.163	<0.001 *	−40.44	−20.73	−1.841	0.073	−40.05	1.64
Thigh Fat area (cm^3^)	13.42 (1.61)	15.9 (1.73)	13.04 (1.74)	14.67 (2.67)	−7.289	<0.001 *	−3.15	−1.80	−2.789	<0.010 *	−2.79	−0.46
Thigh Fat index (%) ^§^	10.23 (1.61)	9.67 (1.21)	8.38 (2.35)	8.41 (2.47)	0.089	0.933	−0.11	0.12	−0.005	0.999	−0.14	0.14
Fat Mass (kg) ^#^	1.74 (0.32)	1.77 (0.32)	1.99 (0.50)	1.95 (0.49)	−0.471	0.642	−0.16	0.10	0.311	0.758	−0.21	0.29
Fat Free Mass (kg)	36.90 (5.39)	51.07 (8.86)	35.19 (6.65)	47.00 (9.67)	−9.360	<0.001 *	−17.18	−11.17	−5.521	<0.001 *	−16.08	−7.53
%FM ^§^	13.71 (2.93)	10.80 (2.98)	18.16 (5.59)	14.19 (5.85)	4.863	<0.001 *	0.22	0.28	2.756	0.009 *	0.15	0.24
Phase Angle ^#^	1.83 (0.15)	1.84 (0.09)	1.77 (0.06)	1.85 (0.21)	−1.121	0.273	−0.08	0.02	−2.161	0.039 *	−0.15	−0.01
R/H (Ω/m)	398.48 (58.30)	318.96 (66.10)	456.54 (93.15)	340.07 (66.54)	7.354	<0.001 *	58.05	101.00	5.831	<0.001 *	76.56	156.38
Xc/H (Ω/m)	66.49 (9.10)	59.72 (6.03)	67.98 (9.61)	61.52 (8.63)	4.381	<0.001 *	3.70	9.84	2.822	0.008 *	1.89	11.03
CMJ test (cm)	28.35 (3.44)	34.43 (4.97)	24.12 (4.89)	28.2 (5.55)	−6.698	<0.001 *	−7.89	−4.28	−3.061	0.007 *	−6.73	−1.42
Sprint 15 m test (s)	2.78 (0.13)	2.47 (0.11)	3.13 (0.19)	2.86 (0.21)	12.711	<0.001 *	0.26	0.36	4.963	<0.001 *	2.91	3.04
RSA 15 x15 m (s)	6.48 (0.23)	5.78 (0.21)	7.07 (0.43)	6.47 (0.46)	14.932	<0.001 *	0.61	0.79	4.691	<0.001 *	0.34	0.87

Note: Bo, Bologna F.C.; Ru, Russi U.S.; SD, Standard Deviation; *t*, student *t*; *p*, *p*-value; circum., circumference; rel., relaxed; cont., contracted; diamet., diameter; SK, skinfold; CMJ, counter movement jump; RSA, repeated sprint ability; SD, standard deviation; *, statistically significant; ∆, difference between; ^#^, logarithmic scale; ^§^, proportion analysis with the *Z*-test; C. I., Confidence Interval.

## Data Availability

The data presented in this study are available on request from the first author.

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
