# Peer review of "Assessment of Body Composition and Physical Performance of Young Soccer Players: Differences According to the Competitive Level"

_biology, 2022, doi:10.3390/biology11060823_

Round 1

Reviewer 1 Report

Please refer to the attached document.

Reviewer 2 Report

Dear authors,

Thanks for the opportunity to review this study. The main strength is regarding the evaluation of athletes of different ages and sports categories. The study design is coherent only with the main outcome, but not with the second outcome.  The analysis discussion tends to be limited, and the results are novel but limited by the study design. Comparisons between clubs are interesting, although there are baseline differences in athletes that are not resolved by the statistics analysis (covariates did not were included). I suggest modifying the initial rationale and moderating the scope of the results obtained in the discussion.

Suggestions:

Main: 

This study aims to compare anthropometric, body 24 composition and physical performance between and within four juvenile categories of two (elite 25 and non-elite) soccer teams and investigates factors that better discriminate among two teams. Although the aim is consistent with what is stated in the title, the justification approach leans towards the possible predictors of player scouting.  

Minor:

Line 3.  I suggest removing "in Italy" from the title.

Lines 20 – 29. In my opinion, it is not clear how this study would improve the scouting of young football players; although the study describes relevant anthropometric characteristics, the study design does not allow isolating possible confounding factors such as training hours, diet, and use of food supplements etc.

Lines. 74 -79. I suggest including references.

Lines. 90-101. Do these anthropometric characteristics affect performance in the sports discipline?

Lines 106-107. I have doubts about how this description of anthropometric characteristics could modify technical and competitive decisions.

Line 110-111. I suggest describing the recruitment of subjects and calculating the sample size.

Lines 348-349. This study design does not allow this information to be inferred.

Line 370.  This study does not consider maturity assessments.

Lines 471-477. It is not clear how their results contribute to the scouting process.

Lines 471 – 477. The comparison between the two teams is difficult because they have different contextual characteristics.

Tables and figures: The tables contain too much information; I suggest re-populating the relevance of dates.

***Please, consider including more information as supplementary material.

Reviewer 3 Report

Dear Authors

I reviewed # biology-1676357-v2 entitled "Assessment of Body Composition and Physical Performance of Young Soccer Players: Differences According to the Competitive Level" which you submitted to the biology. I would like to inform you of the consideration of your manuscript as follows.

Minor points

  1. All numeric symbols 'n' should be changed to italics.
  2. All statistic symbols 'P' should be changed to italics.
  3. There must be a space between the number and the symbol. See the example below. On line 36, …13.01±1.15 years) of different competitive levels (elite - n=98 and non-elite - n=64) → On line 36, …13.01 ± 1.15 years) of different competitive levels (elite – n = 98 and non-elite – n = 64)
  4. There are many grammatical errors. In particular, it is necessary to distinguish between uppercase and lowercase letters. For example, "Descriptive statistics (Mean ± Standard Deviation, SD) were…” → "Descriptive statistics (mean ± standard deviation, SD) were…”
  5. There are too many abbreviations, causing confusion. Abbreviations that are not significantly needed need to be limited.
  6. Describe in detail how to measure circumferences and skinfold thicknesses presented in lines 131 to 134. For example, "The circumference is taken at the mid-point between the bony protrusion on the shoulder (acromion) and the point of the elbow (olecranon process). The observer was to locate this mid-point in the non-dominant arm, with the participant's elbow to be flexed 90 degrees with palm facing upwards."
  7. In Table 1, the unit of BMI is omitted.
  8. There is no need to repeat numerical values including statistical values in tables in the text of the results. The text of the research results should be rewritten.
  9. It is common to indicate 't-value' and 'P-value' described in all tables and texts up to the 3rd decimal place.
  10. If there are many numbers in the table and it is complicated, you can present only 't-value' and attach "*", "**", and "***", as a symbol. For example, if 't-value' is 4.860 and 'P-value' is less than or equal to 0.001, it can be expressed as 4.860***.

Major points

This study shows the result of a lot of effort. And it is logically well documented. However, it is thought that correction is necessary because there is an error in the many parts.

  1. The contents described in lines 277 to 280 are not for 'fat mass', but statistical values for 'fat free mass'. These results are difficult to see as simple errors. Please review the details below.

“As regards body composition parameters fat mass showed significant difference between elite and non-elite U13 and U14, while fat free mass did not differ (U12: t = 1.53, P 278 = 0.14; U13: t = -0.53, P = 0.6; U14: t = 1.35, P = 0.18; U15: t = 1.83, P = 0.08). If %F is considered, elite player of all the categories presented significant lower values than non-elite peers.”

  1. In the discussion, it is important to change the abbreviation to full terms as much as possible, and to interpret the results according to the hypothesis so as not to confuse the topic of this paper.

Sincerely,

Reviewer 4 Report

Line 21, seems it should be ...from not to younger players

Line 82, is body composition really related to attitude? Perhaps it is related to commitment to high performance. Attitude goes back to personality and all that research in the 1950s. That does not seem relevant here.

Lines 108-112 or so, it seems you still need to edit these sentences. Is it attending soccer or participating on the team or etc.? They could be attending soccer anywhere.

Line 145, just place this sentence in the paragraph above

Line 156, comma before which or use that

Line 187, this one sentence needs a home

Line 228, what is the reasoning for Roy's largest root? It is very generous compared to the other tests.

Across manuscript, I see P and P and p and the same with t or t, again with D and F

Table 1, it seems you should use .00 for all values so a 0.6 should be 0.60

Line 347, which needs a comma or make it that

Round 2

Reviewer 1 Report

I will not further comment nor revise this manuscript. 

Author Response

Dear reviewer,

Thank you for your effort and willingness to review and improve our manuscript!

Reviewer 2 Report

Thanks for incorporating the suggestions given in the first round.

Author Response

Dear reviewer,

thank you so much for your help in improving our manuscript!

Reviewer 3 Report

Dear Authors

I re-reviewed # biology-1676357-v2 entitled "Assessment of Body Composition and Physical Performance of Young Soccer Players: Differences According to the Competitive Level" which you submitted to the biology. I would like to inform you of the reconsideration of your manuscript as follows.

Minor points

1. All numeric symbols 'n' should be changed to italics. Unchanged. See your manuscript. I highlighted green color.

2. All statistic symbols 'P' should be changed to italics. Unchanged. See your manuscript. I highlighted green color.

3. There must be a space between the number and the symbol. See the example below. On line 36, …13.01±1.15 years) of different competitive levels (elite - n=98 and non-elite - n=64) On line 36, …13.01 ± 1.15 years) of different competitive levels (elite – n = 98 and non-elite – n = 64) Unchanged. See your manuscript. I highlighted green color.

4. There are many grammatical errors. In particular, it is necessary to distinguish between uppercase and lowercase letters. For example, "Descriptive statistics (Mean ± Standard Deviation, SD) were…” "Descriptive statistics (mean ± standard deviation, SD) were…” Unchanged. See your manuscript. I highlighted green color.

5. There are too many abbreviations, causing confusion. Abbreviations that are not significantly needed need to be limited. Unchanged. See your manuscript. You should explain in detail which part was modified.

6. Describe in detail how to measure circumferences and skinfold thicknesses presented in lines 131 to 134. For example, "The circumference is taken at the mid-point between the bony protrusion on the shoulder (acromion) and the point of the elbow (olecranon process). The observer was to locate this mid-point in the non-dominant arm, with the participant's elbow to be flexed 90 degrees with palm facing upwards." Unchanged somewhat.

7. In Table 1, the unit of BMI is omitted. …” Unchanged. See your manuscript. I highlighted green color.

8. There is no need to repeat numerical values including statistical values in tables in the text of the results. The text of the research results should be rewritten. …” Unchanged. See your manuscript. I highlighted green color.

9. It is common to indicate 't-value' and 'P-value' described in all tables and texts up to the 3rd decimal place. …” Unchanged. See your manuscript. I highlighted green color.

----------------------

In general, many parts have been corrected.
However, the inside of the table must be modified.
In all tables, t-value and p-value must be entered up to 3 decimal places. Otherwise, it's OK to delete the t-value.
One final piece of advice for readers, if they see your paper and don't understand it, as the problem can be serious. 

Author Response

Reviewer 3

In general, many parts have been corrected.
However, the inside of the table must be modified.
In all tables, t-value and p-value must be entered up to 3 decimal places. Otherwise, it's OK to delete the t-value.
One final piece of advice for readers, if they see your paper and don't understand it, as the problem can be serious. 

Authors' reply

Thank you for your suggestion. Although we understand tables are clear with no student t-test columns, we retain these are crucial statistical information. However, we corrected the decimals as you suggested.